# Nitrosative Stress and Its Association with Cardiometabolic Disorders

**DOI:** 10.3390/molecules25112555

**Published:** 2020-05-31

**Authors:** Israel Pérez-Torres, Linaloe Manzano-Pech, María Esther Rubio-Ruíz, María Elena Soto, Verónica Guarner-Lans

**Affiliations:** 1Vascular Biomedicine Department, Instituto Nacional de Cardiología “Ignacio Chávez”, Juan Badiano 1, Sección XVI, Tlalpan, México City 14080, Mexico; loe_mana@hotmail.com; 2Physiology Department, Instituto Nacional de Cardiología “Ignacio Chávez”, Juan Badiano 1, Sección XVI, Tlalpan, México City 14080, Mexico; esther.rubio@cardiologia.org.mx; 3Immunology Department, Instituto Nacional de Cardiología “Ignacio Chávez”, Juan Badiano 1, Sección XVI, Tlalpan, México City 14080, Mexico; mesoto50@hotmail.com

**Keywords:** nitric oxide, peroxynitrite, nitrosative stress, uncoupled NOS isoforms

## Abstract

Reactive nitrogen species (RNS) are formed when there is an abnormal increase in the level of nitric oxide (NO) produced by the inducible nitric oxide synthase (iNOS) and/or by the uncoupled endothelial nitric oxide synthase (eNOS). The presence of high concentrations of superoxide anions (O_2_^−^) is also necessary for their formation. RNS react three times faster than O_2_^−^ with other molecules and have a longer mean half life. They cause irreversible damage to cell membranes, proteins, mitochondria, the endoplasmic reticulum, nucleic acids and enzymes, altering their activity and leading to necrosis and to cell death. Although nitrogen species are important in the redox imbalance, this review focuses on the alterations caused by the RNS in the cellular redox system that are associated with cardiometabolic diseases. Currently, nitrosative stress (NSS) is implied in the pathogenesis of many diseases. The mechanisms that produce damage remain poorly understood. In this paper, we summarize the current knowledge on the participation of NSS in the pathology of cardiometabolic diseases and their possible mechanisms of action. This information might be useful for the future proposal of anti-NSS therapies for cardiometabolic diseases.

## 1. Introduction

Reactive nitrogen species (RNS) include peroxynitrite (ONOO^−^), nitrogen dioxide (•NO_2_), peroxynitrous acid (HNO_3_), dinitrogen trioxide (N_2_O_3_), nitroxyl (HNO), peroxynitrous acid (ONOOH), peroxynitrate (O_2_NOO^−^), peroxynitric acid (O_2_NOOH), nitrosonium cation (NO^+^), nitrate (NO_3_^−^), nitrite (NO_2_^−^) and nitroxyl anion (NO^−^) and can lead to nitrosative stress (NSS) [1,2]. In normal conditions, RNS serve as important intermediaries in cellular physiology and are generated by multiple biochemical processes [2]. They form part of the immune responses serving as nonspecific defenses, and they participate as second messengers in signal transduction pathways. However, RNS increase several toxic molecules and may exacerbate cellular damage in an oxidative stress (OS) environment when they are present in high concentrations [3].

RNS are secondary metabolites of the oxidation of nitric oxide (NO), and their presence is associated with its overproduction by the inducible nitric oxide synthase (iNOS) or uncoupled the endothelial nitric oxide synthase (eNOS) [1,3]. RNS disrupt the cellular redox balance, together with OS. NO at low concentrations may protect cells from proapoptotic effects; however, the exposure to high NO levels induces apoptosis [4].

Damage by NSS can be significant, since RNS are stable substances that easily diffuse into the intracellular organelles and react at extremely elevated speed rates [5]. NSS represents a pathological condition that contributes to the deterioration of organs and systems. It is associated to several cardiometabolic pathologies that include atherosclerosis, hypertension, endothelial dysfunction and diabetes, among others. [2,4,5]. The participation of NSS in the pathophysiology of cardiometabolic diseases may be as important as that played by OS. However, it has been much less described. Therefore, one of the aims of this review is to explore the participation of NSS in these diseases.

Although antioxidant therapies have been proposed for cardiometabolic diseases, the removal of too many ROS and their derived products by antioxidant supplementation may upset the cell signaling pathways and increase the risk of chronic disease. Therefore, there is clinical evidence of the posible failure of antioxidant treatments for OS-associated pathologies such as cardiovascular disease, coronary artery disease and hypertension [6]. The effects of the treatments might depend on the level and type of ROS, the time of exposure and the antioxidant status of tissues.

The possible use of anti-NSS therapies has been proposed for some diseases. NO may potentially improve treatment of cancer, and anti-NSS agents might mitigate toxicity and reduce the required dose for chemotherapy. Nevertheless, scavengers of RNS have also been shown to decrease the activity of some chemotherapy drugs. The possiblilities of the employment of anti-NSS therapies for cardiometabolic diseases have not been described, and additional research on the role of RNS in these diseases is needed to propose new therapies aiming in this direction. Although it is currently recognized that NSS participates in the pathological processes of these diseases, its mechanisms of action are still being elucidated. Here, we discuss the main characteristics of NSS and its role in several pathologies, as well as its mechanisms of action. This information would be needed for the proposal of new therapies.

## 2. NO, Its Synthesis and the Generation of RNS

NO is the main precursor of RNS. It is a multifunctional molecule that participates in signaling cascades via the activation of the soluble guanylate cyclase. Physiological nanomolar (nM) or picomolar (pM) concentrations of NO maintain neurotransmission in the central nervous system, including the retina [7]. NO at these concentrations also participates in the equilibrium in host cell defenses and in vasodilatation determining blood flow to tissues such as the brain, liver, lung, kidney and heart in response to local environmental changes [8]. However, it may also modify proteins through nitrosylation altering their activities. NO is a colorless gas with a low water solubility. It is paramagnetic and decomposes quickly in oxygenated solutions, having a half-life of only a few seconds. It is hydrophobic, which is important for its physiological functions, allowing it to pass freely across cell boundaries [9].

NO is synthesized by the isoforms of NOS, which include the neuronal (nNOS), endothelial (eNOS) and inducible (iNOS) isoforms. The substrate of these enzymes is the amino acid l-arginine (L-arg), and the reaction requires molecular O_2_, flavin adenine dinucleotide, flavin mononucleotide, Ca^2+^, calmodulin, (6R)-5,6,7,8-tetrahydro-L-biopterin (BH_4_) and NAD(P)H as cofactors [10] (Figure 1). The conversion of L-arg to l-citrulline with the production of NO takes place in two steps: the first step is the hydroxylation of l-arginine to N^G^-hydroxy-l-arginine, which is an intermediate metabolite [11], and the second step involves an electron oxidation of the N^G^-hydroxy-l-arginine that binds to two moles of O_2_ and 1.5 mol of reducing equivalent, NAD(P)H, resulting in the formation of a mole of NO. Both steps are catalyzed by the heme iron complex present in the enzyme [9]. The O_2_^−^ required may come from the electron transport chain in the mitochondria, from peroxisomes, NAD(P)H oxidases, xanthine oxidases (ORX), cytochrome p450 and/or the cyclooxygenase (COX) pathways [12].

## 3. ONOO^−^ and NO

Out of all RNS, ONOO^−^ is the most abundant, and its synthesis is related to NO production. NO, in the presence of O_2_^−^, can form all of the RNS, but the predominant molecule formed is ONOO^−^, which is the most cytotoxic [13]. The formation of ONOO^−^ is approximately three times faster than the dismutation of O_2_^−^ by superoxide dismutase (SOD) [14]. The ONOO^−^ decomposition produces •NO_2_, which is also a strong oxidant. •NO_2_ can readily modify a variety of biomolecules, including the aminoacid tyrosine, forming 3-nitrotyrosine and catecholamines. Oxidation of NO_2_^−^ also forms •NO_2_ [15]. Under physiological conditions, ONOO^−^ rapidly decays to its protonated form (pKa 6.8) peroxynitrous acid (HOONO), with a half-life of approximately 1 s. ONOO^−^ is relatively stable under strongly alkaline conditions, but it is, in itself, a reactive species, being able to oxidize sulfhydryl groups 1000 times faster than H_2_O_2_ [16].

Experimental studies in the synthetic system showed that the production rate ratios of O_2_^−^ to NO and/or NO to O_2_^−^ (Q_O2_^−^/Q_NO_ or Q_NO_/Q_O2_^−^) is of one yield maximum ONOO^−^ concentration, which has been associated with the maximum protein tyrosine nitration. These production rate ratios are similar in endothelial dysfunction and in iNOS-induced NO overproduction. This suggests that the ONOO^−^ molecule is highly toxic and is responsible for the NSS in the cellular components, producing macromolecular modifications that contribute to the alteration of normal functions [17].

ONOO^−^ is a product whose formation is favored under conditions such as hyperglycemia, where the cellular production of NO and reactive oxygen species (ROS) are increased. It can be further protonated to produce HNO_3_, which, in turn, can yield the radical hydroxyl (OH^−^) [8]. However, ONOO^−^ formation is usually prevented by antioxidant mechanisms in the cell, such as glutathione (GSH) and/or SOD [18].

RNS production depends on eNOS, and, in turn, eNOS activity is influenced by ONOO^−^. The activity of both eNOS and iNOS can be inactivated via S-nitrosylation by RNS [19]. This was demonstrated when the eNOS activity was decreased by adding the vascular endothelial growth factor to endothelial cells [20]. ONOO^−^ also alters other proteins involved in vascular regulation, including COX-1, mitochondrial enzymes, soluble guanylate cyclase and smooth muscle contractile proteins [21].

Vascular regulation in the cerebral circulation can be impaired by inducing NSS damage to critical proteins. ONOO^−^ may produce nitration in the presence of NO_2_^−^ and hydrogen peroxide (H_2_O_2_) in heme peroxidase enzymes, such as myeloperoxidase, lactoperoxidase and prostaglandin H synthase [22].

NSS may also cause deficiencies in the cofactors needed for the eNOS and nNOS enzymatic reactions, leading to the uncoupling of these enzymes. ONOO^−^ uncouples eNOS and nNOS by direct oxidation on BH_4_ [23]. All NOS enzymes use a BH_4_ cofactor, which is an allosteric activator of NOS and is necessary for the catalytic turnover of the enzyme. In the presence of an overproduction of ROS and in the absence of BH_4_, the electron flow is diverted from the prosthetic heme of the oxygenase domain to molecular O_2_ rather than to L-arg, causing O_2_^−^ production instead of NO [24]. This phenomenon is termed eNOS uncoupling and is partially prevented by the antioxidant, ascorbic acid, which recycles the BH_3_·radical back to BH_4_. This was demonstrated in endothelial cells that were previously treated with both BH_4_ and ascorbic acid, preventing the uncoupling of eNOS by ONOO^−^ [25]. ONOO^−^ oxidizes BH_4_ to the BH_3_ radical, which, in turn, is oxidized to BH_2_^−^_._ The BH_2_^−^ may then compete with BH_4_ for the binding sites at the NOS isoforms, causing a positive feedback loop in the uncoupled enzymes, contributing to NSS formation. Another process that results in uncoupled NOS isoforms is a loss of the reducing equivalents from the reductase region of the enzyme, which allows for the NAD(P)H oxidase to work in the absence of L-arg. This reaction is inversely dependent on the amount of bound BH_4_ and results in the release of O_2_^−^ [26] (Figure 1).

BH_4_ biosynthesis is mediated by the enzyme guanosine triphosphate cyclohydrolase I, which is rate-limiting for its synthesis. Guanosine triphosphate cyclohydrolase I transform guanosine triphosphate (GTP) to dihydroneopterin triphosphate, which is then converted to 6-pyruvoyl-tetrahydropterin by 6-pyruvoyltetrahydropterin synthase. This is a second unstable intermediate compound, which is converted to BH_4_ [27]. An increase or decrease of the guanosine triphosphate cyclohydrolase I activity can lead to subsaturating levels of BH_4_, which result in a deficiency of the cofactor. This alters the electron transfer in NOS that leads to its decoupling, with subsequently less generation of NO and more production of O_2_^−^ [28]. In addition, guanosine triphosphate cyclohydrolase I overexpression in endothelial cells result in an increase of the BH_4_ levels and a decrease in O_2_^−^ generation in hypertrophic myocytes. However, this does not prevent the uncoupling of the eNOS and its associated chronic nonischemic cardiomyopathy [29].

The uncoupled NOS isoforms are one of the in vivo pathways for the generation of HNO. HNO is a one-electron product of the reduction of NO that displays a distinct chemistry from that of NO [30]. HNO and NO exist in chemical equilibrium. HNO may be produced by direct oxidation of N^G^-hydroxy-l-arginine, which is an intermediate specie generated by the uncoupled NOS. HNO can be converted to NO through the reaction of the Fe II molecule at the heme site of the enzyme or by SOD. The NO reduction by orexin and the NO dismutation through catalysis of iron-sulfur complexes also produce HNO. However, these processes oxidize thiols by different reaction mechanisms such as S-nitrosylation versus a disulfide bond formation [31]. In cardiomyocytes, HNO improves contraction and relaxation by activating the ryanodine receptor. This accelerates the influx of extracellular Ca^2+^ ions and increases reuptake by SERCA2a having a positive inotropic effect [32]. The inotropic effects are achieved through the reaction of HNO with specific thiols on key proteins that participate in the the development of the excitation-contraction-coupling machinery. In addition, HNO produces a disulfide bond that alters the conformation of the regulatory protein, thus relieving the inhibition of the Ca^2+^ pump. Other mechanisms by which uncoupled NOS isoforms may act remain unclear and may be related to the depletion of the amino acid L-arg or the accumulation of endogenous methyl-L-arg [33].

On the other hand, iNOS—that is, the physiologically Ca^2+^ independent from the isoenzyme—is usually absent in cells under physiological conditions but is expressed within several hours after the stimulation of cells by inflammatory signals such as interleukins, endotoxin, fever, lipopolysaccharide, sepsis and anaphylactic shock. The iNOS expression is regulated by both transcriptional and post-transcriptional mechanisms, including epigenetic factors. Transcriptional regulation involves the activation of the nuclear transcription factor (NF-ĸB). Figure 1 illustrates the synthesis and the generation of RNS [34].

## 4. Mechanisms of Damage by RNS

The mechanisms by which RNS produce damage remain poorly understood and are discussed in this section. ONOO^−^ may cause lipid peroxidation when the antioxidant mechanisms fail. Among the products of lipid peroxidation malondialdehyde, 4-hydroxinonenal and conjugated diene may be mentioned [1,35]. ONOO^−^ can cause the nitration of tyrosine residues in intracellular proteins. The nitration of tyrosines involves incorporation of the ONOO^−^ and •NO_2_ production by heme proteins [36]. The reaction between H_2_O_2_ and NO_2_^−^, with a tyrosine residue in the ortho-position of the phenolic hydroxyl group, results in the formation of 3-nitrotyrosine. 3-nitrotyrosine may damage proteins or render them less active. Nitrotyrosine is an index of RNS formation [37]. An example of the effect of nitration in proteins is the modification of eNOS by this reaction, reducing the bioavailability of NO and attenuating NO-dependent responses mainly in the endothelium and blood vessels [38].

ONOO^−^ can also interact with proteins via S-nitrosylation at specific cysteine residues to alter their function. Nitrated cysteines are the primary modification involved in redox signaling events in the cell. Cysteine residues are present in most proteins, and cysteine is the second-most abundant amino acid in them (approximately 1.9% of total amino acid composition) [39]. However, only a small percentage of cysteine residues within proteins are susceptible to modification by ONOO^−^. The polar and hydrophilic thiol groups of cysteine in protein side chains are susceptible to oxidation by ONOO^−^ to give rise to disulfide bridges, resulting in increased protein stiffness and proteolytic resistance. In this regard, there are conserved cysteine residues in almost all proteins that are often critical for protein functioning [40].

S-nitrosylation is a nonenzymatic reversible reaction that consists of the covalent attachment of NO to a reactive cysteine residue to form S-nitrosothiols, which include low-molecular-weight S-nitrosoglutathione and S-nitrosylated proteins. The S-nitrosylation can occur through transnitrosylation involving an acceptor thiol and S-nitrosoglutathione [41]. The S-nitrosoglutathione is the most abundant endogenous S-nitrosothiols, and it serves as a stable intracellular NO reservoir. A key mechanism for regulating the S-nitrosothiol action involves the enzymatic activity of the alcohol dehydrogenase III, also known as S-nitrosoglutathione reductase [42]. S-nitrosoglutathione reductase selectively metabolizes S-nitrosoglutathione and indirectly depletes the *S*-nitrosylated protein levels, which are in dynamic equilibrium with S-nitrosoglutathione. Deficiency of the S-nitrosoglutathione reductase activity results in high S-nitrosoglutathione levels and *S*-nitrosylated proteins [43]. S-nitrosoglutathione reductase activity changes affect the entire S-nitrosothiols pool and can modulate cellular signaling.

Furthermore, S-nitrosoglutathione reductase deficiency has been associated with NSS and impaired cardiovascular function [44], tissue damage and increased mortality in mouse models of sepsis. This has been associated with GAPDH S-nitrosylation, which results in the subsequent covalent inactivation of the enzyme [45]. This irreversible modification requires the synthesis of new proteins to restore GAPDH activity in the cells. The S-nitrosothiols sensitize endothelial cells to redox-cycling bioenergetic dysfunction and favors death with the concomitant elevation of cellular S-nitrosothiols levels [46].

## 5. Mitochondrial Proteins and DNA

On the other hand, the mitochondria are responsible for the production of energy in the form of adenosine triphosphate (ATP), which is used by cells for their functioning and survival [47], and the functioning of these organelles is altered by NSS. The function of mitochondria is especially important in the heart due to the high demands of energy. This energy is provided by oxidative phosphorylation. Defects in the activity of the electron transport chain and in the oxidation of fatty acids in mitochondria have also been involved in the pathogenesis of type 2 diabetes and other metabolic diseases. Additionally, the mitochondria play a critical role in the antioxidant defenses, since they contain SOD isoforms, catalase, thioredoxin 2, glutathione peroxidase 1 and 4, glutathione and α-keto acids in their matrix. These enzymes detoxify the hydrogen peroxide generated in the cytoplasm, which is then imported into the mitochondrial matrix by the peroxiporin [48]. Mitochondria also participate in the β-oxidation of fatty acids; in apoptosis and in the intermediary metabolism (such as the glutamine metabolism, urea, pyrimidine, ammonia and steroid metabolism, among others).

ONOO^−^ oxidatively inactivates different mitochondrial proteins, including electron transport chain components and citric acid cycle dehydrogenases [49]. Mitochondrial proteins and DNA are more susceptible to NSS damage than their nuclear counterparts due to the absence of protective catalase, histones or polyamines. Moreover, there is a relatively low activity of DNA reparation enzymes in mitochondria when compared to their amount in the nuclei. The mutations and/or deletions of the mitochondrial DNA through nitrosatives modifications are important, since mitochondrial DNA encodes for 13 polypeptides, all of which are subunits of the complexes I, III, IV and ATP synthase (complex V) of the mitochondrial electron transport chain [50]. The damage induced by nitration was demonstrated in the alcoholic fatty liver from rats, where the complex V was significantly inhibited through the nitration of Tyr residues in the catalytic β subunit of the ATP synthase [51]. This nitration of the catalytic subunit of ATP synthase could contribute to its inactivation and lead to decreased ATP levels. In addition, another study demonstrated that both nonalcoholic fatty liver and alcoholic diseases may result from impaired mitochondrial functions with suppressed β-oxidation of fatty acids [52].

The ONOO^−^ can also covalently modify proteins through the S-nitrosylation of Tyr and Cys residues in the mitochondria. Tyr nitration is the major mechanism responsible for the irreversible inhibition of mitochondrial complex I provoked by the addition of exogenous NO that increases ONOO^−^ levels [53]. In the mitochondria isolated from rat hearts, ONOO^−^ impairs complex I by S-nitrosylation through damage of the iron-sulphur centers of complex I. However, the S-nitrosylation of protein thiols can be reversed by reduced thiols through GSH [54]. Figure 2 describes the relationship between NSS and mitochondrial proteins.

ONOO^−^ also interacts with nucleic acids, and, in this reaction, 8-hydroxydeoxyguanosine and 8-nitroguanidine are formed, which can cause breaks and single-strand formation in DNA. Studies in vitro have demonstrated that both endogenously and exogenously generated ONOO^−^ lead to the overactivation of the nuclear enzyme poly-(ADP-ribose) polymerase (PARP-1) [55]. PARP-1 contributes to DNA reparation and to the maintenance of genomic stability. This overactivation consumes nicotinamide adenine dinucleotide (NAD^+^) and, consequently, ATP. ONOO^−^ may activate PARP-1, and this may be the major mechanism of DNA injury in circumstances associated with NSS, which in turn, depletes cellular NAD^+^ and ATP, promoting cellular dysfunction and necrosis [56].

The DNA single-strand breakage is the obligatory trigger for the activation of PARP-1. PARP-1 uses NAD^+^ as substrate to form poly-ADP-ribose and plays a role in numerous physiological mechanisms such as DNA repair, the regulation of genomic stability and gene expression. This enzyme is also able to PARylate itself in an automodification reaction. The PARP-1 is the most abundant isoform of the PARP enzyme family [57]. The PARP-1 is a zinc finger protein that belongs to a family of 18 identified genes and transcribes PARP enzymes that catalyze the covalent transfer of poly-ADP units from NAD^+^ to acceptor proteins. The PARP-1 has three functional domains: (1) a DNA-binding domain that contains a nuclear localization signal, (2) an automodification domain that acts as an acceptor for poly-ADP-ribose units and (3) a C terminus catalytic domain [58]. In normal conditions, PARP-1 participates in DNA base excision repair and in the maintenance of the genomic stability. However, it can be overactivated by ROS and RNS to induce DNA damage. It then rapidly uses the substrate β-NAD^+^ to transfer poly-ADP-ribose to itself and to nuclear acceptor proteins. In an effort to resynthesize NAD^+^, the cell consumes its ATP pools and reaches an energy crisis, resulting in cell death. Upon binding to damaged DNA, PARP-1 forms homodimers and catalyzes the cleavage of NAD^+^ into nicotinamide and ADP-ribose to form ADP-ribose polymers, long branches on glutamic acid residues of a number of target proteins, including histones, and by PARP-1 automodification. NSS triggers extensive DNA breakage, PARP-1 overactivation and the consequent depletion of the cellular stores of substrates such as NAD^+^, impairing the Krebs cycle, glycolysis and mitochondrial electron transport. This results in ATP depletion and the consequent cell dysfunction and death by necrosis [59].

The PARP pharmacological inhibition or genetic deletion preserves cellular NAD^+^. ATP pools in oxidatively and/or nitrosatively stressed endothelial cells (as well as many other cell types) and allows them to function normally. When the apoptotic process has initiated, these molecules are utilized by the apoptotic machinery and are used for cell death [60]. ONOO^−^ causes DNA single-strand breakage and activation of the nuclear enzyme PARP-1 in endothelial cells placed in high extracellular glucose concentrations and in the blood vessels of diabetic rodents. This activation by ONOO^−^ of the PARP-1 axis plays an important role in myocardial ischemia, diabetes and diabetes associated with cardiovascular dysfunction [61].

## 6. Connections with Antioxidant Enzymes

The maintenance of cellular redox homeostasis in which antioxidant enzymes participate is important for the functioning of the various metabolic pathways in the cell, and NSS disrupts it. Several antioxidant enzymes are altered by ONOO^−^. Catalase may undergo several post-translational modifications that impact its activity, including nitration and S-nitrosylation. These modifications inhibit the active center of catalase after exposition to a NO overconcentration by inducing changes in the heme group biosynthesis and degradation [62]. CAT is an oxido-reductase enzyme that catalyzes the decomposition of H_2_O_2_ into molecular O_2_ and H_2_O through its heme group. It uses manganese (Mn^2+^) as a cofactor. Catalase helps in protecting and detoxifying cells against damage by an H_2_O_2_ overproduction [63]. Catalase has a heme-tetramer that binds four Fe-protoporphyrin IX to it and two-to-four NAD(P)H molecules. The heme groups are deeply buried in the structure using tyrosine, asparagine and histidine as ligands. Thin channels provide access of H_2_O_2_ to the heme group [64]. This interaction of the heme groups with ONOO^−^ results in ferric-nitrosyl formation and metallo-oxo, which prevents the binding of H_2_O_2_ to the metal ion, thus inhibiting the activity of catalase [65]. In addition, four NO molecules form a complex with each catalase tetramer. When NO is removed from the catalase binding sites, the full enzymatic activity of the native catalase can be restored. However, high H_2_O_2_ concentrations may promote NO-binding [66].

On other hand, the three isoforms of SOD may also be altered by NSS. SOD isoforms have cofactors such as Cu^2+^, Zn^2+^, Mn^2+^ and Fe^2+^ in their catalytic sites. The Cu/Zn-SOD isoform is located in the cytoplasm, the Mn-SOD is located in mitrochondrial matrix and the Cu/Zn-SOD extracellular isoform is found in plasma [67]. The SOD isoforms are the primary defense of cells against the oxidative injury caused by O_2_^−^ [68]. These enzymes decrease the formation of 3-nitrotyrosine in tissues by their interaction with ONOO^−^. This interaction was first reported for bovine Cu/Zn-SOD on Tyr-108. However, in humans, the nitration occurs in Trp-32 [69]. The Try-nitration in human Mn-SOD result in a decrease of its enzymatic activity with a concomitant increase in Tyr-34, and this is associated with an altered iNOS pathway [70]. The nitration of Tyr-34 in Mn-SOD in the different organs could be due to high NO concentration (5–200 µM) that may diffuse through the mitochondrial membrane. This modification enhances O_2_^−^ production by inhibition of the cytochrome oxidase, which leads to the accumulation of more O_2_^−^ that is not detoxified by Mn-SOD. This accumulation of O_2_^−^, in turn, oxidizes the NO produced by iNOS, and ONOO^−^ is formed, which then accumulates and competes for the common substrate O_2_^−^ [71]. This process causes damage to mitochondrial components and, subsequently, results in mitochondrial dysfunction that leads to apoptosis or cell necrosis [72]. ONOO^−^ may inactivate the Mn-SOD isoform through 3-nitrotyrosine in humans. In this process, a single nitro group is substituted onto in the Tyr34 residue of the catalytic site, resulting in an increase in S-nitrosoglutathione. The S-nitrosylation of the catalytic site in the enzyme may also lead to the loss of its activity and result in a 97% inhibition [73].

The glutathione peroxidase (GPx) isoform family consists of homologous enzymes that are selenium-dependent, since their catalytic site is a seleno-cysteine. The GPx1 is the main antioxidant enzyme preventing the accumulation of intracellular damage by H_2_O_2_. It uses GSH and belongs to the GPxs family. GPx1 also acts as a peroxynitrite reductase to modulate the in vivo ONOO^−^ flux. The lack of GPx1 enhances ONOO^−^ survival [1]. The GPx molecule also contains cysteines that form part of peroxiredoxins that catalyze the reduction of peroxides such as H_2_O_2_, lipid or alkyl hydroperoxides to H_2_O and the corresponding R-OH^−^. The peroxiredoxin sites function either as dimers or monomers and contain one or two Cys residues at their active site. In the case of 2-Cys peroxiredoxin, small oxide-reductases with redox centers comprising two Cys-residues regenerate the catalytic site, whereas a cysteine residue in GPx is reduced by GSH [74]. Peroxiredoxins are capable of detoxifying the ONOO^−^, and they regulate its bioavailability, being targets of S-nitrosylation [75].

Glutathione reductase is the enzyme responsible for GSH regeneration, and its activity controls the oxidized glutathione levels. Glutathione reductase has two thiols present at its active site, making it a potential target for S-nitrosothiols, which can lead to the inhibition of its activity after undergoing S-nitrosylation, causing decreases in the GSH cellular levels. There is a significant inhibition of glutathione reductase activity and a reduced concentration of GSH after exposure of cells to an excess of NO [76]. The reaction of ONOO^−^ with GSH by nitrosylation can form S-nitrosoglutathione, which is involved in a number of physiological processes, including apoptosis and the response to NSS and OS [77]. This process may deplete the total antioxidant capacity, affecting cellular survival [78]. Figure 3 describes the nitration and nitrosylation processes of the antioxidant enzymes that drive cells to a deficient antioxidant defense.

## 7. Inflammatory Process

RNS exacerbate inflammatory processes that are involved in the development of cardiometabolic disorders. Cardiometabolic diseases such as obesity and metabolic syndrome are low-grade systemic inflammatory conditions, and inflammation is a key component of endothelial dysfunction. Inflammation is mediated by secretory agents from adipose tissue and the liver. These agents alter the function of different tissues and organs and contribute to cardiovascular complications.

A variety of mediators, including ROS and RNS, trigger iNOS expression during the inflammatory process. iNOS modulates acute and chronic inflammatory conditions, and NO accumulates in inflamed tissues [79].

The reaction of NO with O_2_^−^ is facilitated in inflammatory conditions, and more NO is produced by the infiltration of phagocytes that generate O_2_^−^. Under these conditions, ONOO^−^ is a more powerful oxidant than O_2_^−^, because the former has a higher diffusion coefficient and a longer half-life [1]. Both O_2_^−^ and ONOO^−^ are involved in initiating, amplifying and propagating tissue damage during inflammatory processes by their ability to modulate the inflammatory interleukin expression, including factors such as IL-6, in the phagocytes [80].

ONOO^−^, in addition to being a direct mediator of cell death and a trigger for a variety of proinflammatory processes, can contribute to the overexpression of molecules that slow down monocyte migration, such as ICAM-1, VCAM and P-selectin in human endothelial cells, contributing to the deterioration of the vascular function. These effects are mediated, in part, by the ability of ONOO^−^ to enhance the NF-ĸB mediated proinflammatory signal transduction pathways. In addition, the GAPDH S-nitrosylation by ONOO^−^ and its subsequent inactivation contributes to activate NF-ĸB. NSS also plays a crucial role in the development of vascular complications such as thoracic aortic aneurysm formation through damage induced by ONOO^−^ [81].

The inflammatory process can also regulate the production of NO through epigenetic mediators. MicroRNAs are small, noncoding RNAs that bind to a specific 3′-untranslated region of mRNA and inhibit its translation or promote its degradation. MicroRNAs are also important in the endothelial function and are associated with hypertension and preeclamsia. Some factors released during the inflammatory processes, such as TNF-α, might decrease eNOS expression and NO production by blocking the functional activity of the eNOS mRNA 3′-untranslated region through the production of the microRNA-31-5p and microRNA-155 [82]. On the other hand, microRNA-212 have been linked to several processes in the brain where they regulate NF-ĸB and nNOS, respectively. Moreover, microRNA-132 inhibition leads to excessive NO production and thereby affects the S-nitrosylation of critical neuronal proteins, leading to neurological diseases [83]. In this paper, we focus on the effects of NSS on cardiometabolic disorders, but changes that occur in these and other pathologies are summarized in Figure 1.

## 8. Metabolic Disorders

### 8.1. Diabetes Mellitus

On the other hand, diabetes mellitus is a chronic metabolic disease that results in hyperglycemia, which causes many complications in organs and systems. Hyperglycemia is an important trigger for RNS formation. NSS in diabetes mellitus may cause hemodynamic alterations through the overexpression of iNOS, which is associated with vascular, cardiac and renal tissue damages in streptozotocin diabetic rats [84]. There are also increased levels in ONOO^−^ in the endothelial cells of human aortas exposed to elevated glucose concentrations and in blood vessels of type II diabetic patients. ONOO^−^ is also increased in other models of experimental diabetes [85].

NSS appears after at least six weeks of diabetes in experimental animals. Oscillating glucose triggers a higher degree of ONOO^−^ generation than stable high glucose in cultured human umbilical endothelial cells [86]. In diabetes mellitus, an increase in the ONOO^−^ production not only causes NSS, but it also reduces the bioavailability of functional NO and contributes to impaired endothelium mediated relaxation. However, the treatment of diabetic rats with N-acetyl cysteine significantly improves blood pressure, endothelial relaxation and heart rate, while decreasing the iNOS expression. The N-acetyl cysteine treatment may reduce the ONOO^−^ formation, possibly by the simultaneous inhibition of ROS and NO via NF-ĸB suppression, which mediates the induction of the expression of iNOS [87].

There are high levels of 3-nitrotyrosine in plasma in diabetes mellitus patients that are associated with myocytes, fibroblasts and endothelial cell apoptosis. The increase of microvascular 3-nitrotyrosine immunoreactivity correlates with the glycemic control and with levels of intracellular and vascular adhesion molecules [88].

Hyperglycemia, which is one of the main signs of diabetes mellitus, induces endothelial cell dysfunction through an increased flux in the polyol pathway, increasing diacylglycerol formation with subsequent protein kinase C (PKC) activation. The increase in blood sugar also induces the formation of advanced glycation end products that cause many of the complications found in the disease. The accelerated nonenzymatic formation of advanced glycation end products in the presence of hyperglycemia increases the expression and activity of NAD(P)H oxidase, which then generates more O_2_^−^ via PKC activation [89]. This increased O_2_^−^ levels, uncoupled eNOS and increased the iNOS activity, with a subsequent increase of the 3-nitrotyrosine levels. Besides, 3-nitrotyrosine formation happens in the walls of the arteries of hyperglycemic monkeys and in diabetic patients during postprandial hyperglycemia. Nitrotyrosine formation is associated with an increase of HbA1c, ICAM and VCAM levels [90].

### 8.2. Diabetic Retinopathy

Retinopathy is the principal cause of visual loss and blindness in diabetic patients. It is characterized by microvascular dysfunction. Hyperglycemia is considered as the major pathogenic factor for the development of diabetic retinopathy, and it is associated with increased OS and NSS in the retina that is associated with apoptosis [91]. Vascular dysfunction in the retina is characterized by an increase in capillary occlusion; increased blood-retinal barrier permeability; the formation of microaneurysms, lipid exudates and cotton-wool spots and the appearance of hemorrhages. At the later stages, neovascularization and macular edema can also occur. These anatomical and structural changes are associated with NSS and result in neural cell death and apoptosis [92].

The anti-inflammatory, antiapoptotic and antioxidant heme oxygenase-1 enzyme is expressed under stress conditions, such as the NSS, and is involved in the protection of retinal cells. The heme oxygenase-1 family is composed of three iso-enzymes: two constitutive ones form the heme oxygenase-2 and -3 and an inducible one forms heme oxygenase-1. These enzymes catalyze the degradation of heme groups into equimolar amounts of biliverdin, ferrous iron and carbon monoxide [93]. Under diabetic conditions, the heme oxygenase-1 protects retinal endothelial cells exposed to elevated glucose or OS and NSS conditions. The inhibition of heme oxygenase-1 increases ROS production, and the toxic effect of these free radicals are induced by the exposure to H_2_O_2_, a NO donor (NOC-18). The heme oxygenase-1 overexpression can prevent the toxic effect-induced NOC-18 in retinal endothelial cells exposed to hyperglycemia and conditions such as OS and NSS [94].

Furthermore, the NO/soluble guanylate cyclase/guanosine cyclic monophosphate (cGMP) pathway is also involved in the modulation of visual information processing in the retina. However, in diabetic retinopathy, this pathway is impaired by S-nitrosylation, leading to visual loss and blindness [95]. The mechanisms initiated by hyperglycemia are not completely understood; however, the relationship between increased aldose reductase activity, NSS and PARP-1 activation has become a focus of interest in the diabetic lens, nerve and retina [96]. The aldose reductase is the first enzyme of the sorbitol pathway in the glucose metabolism, and this enzyme has an increased activity that leads to NSS in diabetic conditions. The increase in the aldose reductase activity is responsible, in part, for the lipid peroxidation, NSS, and the activation of PARP-1 in the diabetic rat and mouse retinas, and it depletes the cells of GSH, ascorbate and taurine [97].

Eales disease is an idiopathic inflammatory venous occlusion that has male predominance (99%). It affects the peripheral retina of young adults (15 to 45 years). Inflammation is the main clinical feature of this pathology, and this is associated with increases in the aldose reductase activity, which catalyzes the NAD(P)H-dependent reduction of glucose to sorbitol. The increase of the sorbitol concentrations leads to decreases of the Na/KATP-ase activity and to an imbalance in the NAD^+^/NADH and NADP^+^/NAD(P)H redox leves. These changes are associated with NSS through the formation of advanced glycation end products, activation of PKC and mitogen-activated protein kinase and an increased aldose reductase activity, which is linked to the destruction of the capillary retinal cells [98].

### 8.3. Obesity

Obesity and other associated pathologies are multifactorial disorders that are linked to the increase of ROS and NSS [99]. Obesity is a low-grade inflammatory condition that contributes to cardiovascular diseases, because adipose tissue produces and releases several proinflammatory mediators such as free fatty acids, interleukins and leptin [100]. This inflammatory condition leads to an increase in the expression of iNOS and, in consequence, to an increase in the production of NO and in the formation of RNS and protein S-nitrosylation [101]. The direct nitrosylation of different target proteins affects, in a concentration-dependent manner, the main metabolic pathways, such as insulin signaling, lipolysis and adipogenesis. S-nitrosylation of the fatty acid synthase, a key enzyme of fatty acid synthesis, is associated with its dimerization, leading to an increase in its activity, thus contributing to adipogenesis [102].

In the same way, lipolysis and β-oxidation are modulated by NO through (1) a direct activation of 5’ adenosine monophosphate-activated protein kinase (AMPK) and (2) the regulation of the activity of the hormone-sensitive lipase. Lipolysis, which is dysregulated in obesity, leads to lipotoxicity, a process that increases the release of free fatty acids and their accumulation in ectopic deposits, such as skeletal muscle and the liver [103].

### 8.4. Fatty Liver Disease

In alcoholic and nonalcoholic fatty liver, there is an increase in NSS [104]. NSS modulates hepatic inflammation, fibrogenesis and the accumulation of triacylglycerols. In addition, an increase of the ROS and RSN impair the secretion of very low-density lipoproteins by reducing the ApoB100 and acyl-coenzyme-A expressions, which then lead to the accumulation of triacylglycerols in hepatocytes [105]. The NSS in hepatic steatosis can eventually result in hepatic insulin resistance, fibrogenesis and inflammation via the activation of C-Jun, the N-terminal kinase and the PKC pathway, thus perpetuating fatty liver. NSS predicts the presence and severity of nonalcoholic fatty liver at different stages of the development of insulin resistance and metabolic syndrome, where vitamin A intake may be playing an important role. The intake of vitamin A and the ingestion of monounsaturated fatty acids may reverse damage to the liver [106].

## 9. Cardiovascular Disorders

NSS plays an important role in the pathogenesis of cardiovascular diseases, including heart failure, ventricular hypertrophy, atrial fibrillation and myocardial ischemia. There is an increase in 3-nitrotyrosine formation in subjects with cardiovascular diseases. Therefore, 3-nitrotyrosine can be considered as a biomarker of RNS in cardiovascular diseases [107].

In cardiomyocytes treated with IL-1 β, IFN-gamma and lipopolysaccharide, there is an increase in the ONOO^−^ levels, which is associated with the overexpression of iNOS. This causes deleterious effects in the myocardial energy balance by attenuation of the myocardial inotropic response to β-adrenergic stimulation and by the induction of cardiomyocite necrosis. In addition, cytosolic iNOS overexpression decreases intracellular Ca^2+^, reduces the Ca2þ spark frequency and depletes ATP in cardiomyocites [108]. This could result in cardiovascular depression, manifested by a decreased heart rate, diminished mean blood pressure and cardiac output and attenuated pressor responses to vasoactive agents in hyperglycemia in animal models of type 1 diabetes. In addition, ONOO^−^ mediates abnormalities in the vascular tissues in diabetes by direct oxidation of catecholamines. This agent also reduces the binding capacity of the α-adrenergic receptors, decreasing the reactivity to vasoactive agents. This suggests that ONOO^−^ may cause cardiac dysfunction and vascular hypo-reactivity, leading to depressed blood pressure. Furthermore, many of the cardiovascular abnormalities in diabetic rat models may be prevented by inhibiting the ONOO^−^ formation in cardiomyocites and endothelial cells [109].

In addition, vascular hyporeactivity is related with an increase in NO that depends on iNOS expression in response to inflammatory interleukins that are associated with bacterial infection and ONOO^−^. This may induce the vascular failure caused by a PARP-1 decrease [110] that contributes to endothelial dysfunction. In vitro, DNA damage and PARP-1 activation occur in endothelial cells exposed to ROS [111]. The ONOO^−^ infusion in isolated perfused hearts results in severe impairment of the endothelial dependent relaxation. In vivo, the formation of ONOO^−^ is detectable in aortas isolated from endotoxemic rats. It is also formed in models of hemorrhagic shock, which triggers the expression of inflammatory mediators and causes increases in ROS production that contributes to the circulatory failure. In addition, ONOO^−^ inhibits the mitochondrial respiratory chain and triggers apoptosis at the subcellular level in cardiomyocytes [112].

In the formation of the atherosclerotic plaque, there is an increase in RNS levels that facilitates the infiltration by monocytes and macrophages and causes damage to cell membranes and DNA in endothelial cells [113]. The interactions between leukocytes and the endothelial cell and the activation of leukocytes are important for the onset and progression of atherosclerosis and for other vascular diseases, including cardiomyopathy, nephropathy, neuropathy, retinopathy and angiopathy. These interactions cause a significant increase in ONOO^−^ and tyrosine nitration in different regions of the capillary microvasculature [114]. This interaction is also associated with increases in Ca^2+^ levels in the endothelial cell that initiate signaling pathways that increase vascular permeability. The increase of the vascular permeability may cause leukocyte extravasation into the tissular region, leading to tissue injury and complications such as edema [115]. In addition, the leukocyte extravasation can increase the ONOO^−^ concentration within leukocytes and can prime them by upregulating the receptors to secondary stimulus on their surfaces by depolarizing their mitochondrial membrane, by increasing the actin polymerization, by enhancing the NAD(P)H oxidase activity and by increasing the intracellular Ca^2+^ concentration [116].

In preclinical models of ischemia-reperfusion injury, the administration of S-nitrosothiols such S-nitrosocysteine and mitochondrial-targeted S-nitroso-N-acetylpenicillamine improve the functional recovery of the heart upon reperfusion [117]. Protection from ischemia-reperfusion injury occurs through the S-nitrosylation of several metabolic enzymes, including glyceraldehyde-3-phosphate dehydrogenase and the complex I of the mitochondrial electron transport chain. Nitrosylation of these proteins inhibits their activity and mitigates oxidant production during reperfusion [118].

In addition, in patients undergoing coronary bypass surgery, there is a three-fold increase in ONOO^−^ formation and a 90% decrease in O_2_^−^ production in the implanted mammary arteries after the addition of sodium nitroprusside, a donor of NO, which impacts the vascular reactivity. Furthermore, a NO imbalance also regulates muscarinic communication in the heart and arteries. The release of acetylcholine in arteries and vagal bradycardia are facilitated by NO. When sodium nitroprusside, a NO donor, is administrated and field stimulation is applied to hearts, it increases the release of acetylcholine and the heart rate response to vagal nerve stimulation. This is due to the stimulation of soluble guanylyl cyclase and an increase in intracellular cGMP [119].

### 9.1. Endothelial Dysfunction

The production of NO via eNOS is the main regulatory pathway of the vascular system, and eNOS is required for the maintenance of the basal vascular tone, the control of permeability, the prevention of smooth muscle cell proliferation, the regulation of vasodilation and the prevention of leukocyte adhesion and/or platelet aggregation. The participation of eNOS in the endothelial function is demonstrated by endothelium-dependent relaxant responses to vasoactive mediators such as bradykinin, acetylcholine and ATP [120]. The lack of NO bioavailability by the uncoupling of eNOS is the main mechanism of endothelial dysfunction. The degree of uncoupling of eNOS is represented by the ratio of the reduced form of BH_4_ to its oxidized form [121].

Endothelial dysfunction induces RNS production from sources other than eNOS, such as NAD(P)H oxidase and mitochondria. The participation of both NO and ONOO^−^ in the endothelial cell function is regulated by their synthesis rate, liberation and their paracrine effects on the vascular smooth muscle cells [122]. ONOO^−^ is increased through its decomposition products •NO_2_, carbonate radical (CO_3_^−^) and OH that cause multiple deleterious effects in endothelial cells. These include the tyrosine nitration of proteins that increase vasoconstriction and reduce vasodilatation. SOD inactivation also increases NSS and induces tissue damage [123]. Studies in synthetic experimental systems have shown that ONOO^−^ formation is strongly dependent on the O_2_^−^ and NO production rates. Their rates of production indicate the relation of NSS to endothelial dysfunction and/or to the overexpression of iNOS [124].

An excess of NO is oxidized to ONOO^−^, which initiates multiple pathological signaling events, including the elevation of interleukins, VCAM, ICAM, P-sectins, PKC, mitogen-activated protein kinase signaling pathways, PARP-1 and NFκB. These molecules may contribute to a positive feedback in an acute inflammatory state, which impacts on smooth muscle cells and can contribute to the inhibition of vasodilation and to an increase in vasoconstriction [125]. They may act through inactivation of the soluble guanylate cyclase, the NO intracellular receptor within the cytosol of the vascular smooth muscle cells [126]. The soluble guanylate cyclase is the main NO receptor and is responsible for the enzymatic conversion of GTP into cGMP. However, the activity of soluble guanylate cyclase may be impaired by S-nitrosylation, leading to a decrease in vascular reactivity. The S-nitrosylation of soluble guanylate cyclase causes a desensitization of NO stimulation [127]. In this sense, both in vivo and in vitro studies have demonstrated that the soluble guanylate cyclase nitrosylation in Cys 516 of the α-subunit correlates with the NO resistance syndrome by angiotensin II, which induces hypertension [128].

In addition to the altered formation of NO in NSS, there are prooxidant effects on the vascular endothelium through the formation of ONOO^−^, which may lead to an impairment of vascular relaxation induced by angiotensin II. The formation of reactive species is mediated, in part, by some NAD(P)H oxidase subunits and nonphagocytic NAD(P)H oxidase proteins [129]. Angiotensin II activates these NAD(P)H oxidase subunits and results in ROS formation that exerts direct oxidative effects on metabolic pathways such as mitogen-activated protein kinase, tyrosine kinases and the synthesis of transcription factors. This may lead to inflammation, hypertrophy, remodeling and angiogenesis [130].

Furthermore, angiotensin II can induce PARP-1 activation, protein 3-nitrotyrosine formation, eNOS uncoupling, BH_4_ reduction and DNA breakage through this pathway. The angiotensin II-PARP-1 pathway has not only been described in endothelial dysfunction but, also, in human diabetes and in a rat model with essential hypertension [131]. There is also evidence that angiotensin II may induce NSS in peripheral organs. In addition, angiotensin II can directly activate NF-ĸB and/or indirectly activate it through the production of O_2_^−^, which leads to iNOS induction, resulting in NO and ONOO^−^ overproduction that result in BH_4_ oxidation and leads to the uncoupling of eNOS. However, BH_4_ supplementations might reduce the vascular damage caused by angiotensin II, thereby preventing the uncoupling of eNOS and decreasing the NSS. The BH_4_ has been shown to scavenge O_2_^−^ with a constant rate of 105 moles per second and may react with ONOO^−^ [132].

### 9.2. Essential Arterial Hypertension and Metabolic Syndrome-Related Hypertension

The generation of RNS also plays an important role in the etiology of metabolic syndrome-related and diabetes-related hypertension. Damage produced by OS and NSS in the brain, vasculatures, kidneys and blood vessels participates in the generation of hypertension in the metabolic syndrome [133].

Hypertension is an important risk factor for coronary heart disease, cerebrovascular accidents, the development of aortic distension and renal failure. In the general population, about 25% of individuals are hypertensive, and the prevalence increases with age. Hypertension affects the endothelial function and the structure of blood vessels. Vessels are formed by three layers: the intima, media and adventitia, which contribute to the increase of the blood pressure [134].

The alterations of the media result in the remodeling of the blood vessel walls, which contributes to vascular smooth muscle hypertrophy and hyperplasia, which, importantly, participate in chronic hypertension. Vascular wall hypertrophy and endothelial damage may cause the reduction of the bioavailability of NO, which contributes to an increase in the vascular resistance and the contractile effect of vasoactive agonists. This reduction is also associated to an increase of the O_2_^−^ and subsequent formation of the ONOO^−^ [135]. Figure 4 describes the relationship between NSS and metabolic disorders that are addressed in this review.

In spontaneously hypertensive rats, endothelial dysfunction is associated with an increase in uncoupled eNOS and with the generation of ONOO^−^. In other rodent models of hypertension, NAD(P)H oxidase and low levels of BH_4_ contribute to OS, which uncouple eNOS and cause the increase in the blood pressure [136].

The eNOS in the aorta of hypertensive animals is also uncoupled by an increase of the COX-2 activity and by a decrease in the amino acid L-arg. This has been described in Dahl-salt-sensitive rats, resulting in an accelerated sodium reabsorption, which contributes to hypertension. However, treatments with L-arg in spontaneously hypertensive rats reduce vascular smooth muscle cell hypertrophy and enhance the NO-cGMP pathway [137]. Another study demonstrated that N-acetyl cysteine protects the vascular function in spontaneously hypertensive rats from NSS and reduces the Glutathione disulfite (GSSH)/GSH ratio, lipid peroxidation and 3-nitrotyrosine levels [138].

In addition, the cholinergic system plays an important role in regulating the vascular tone by stimulating the production of NO. Cholinergic transmission involves the release of the neurotransmitter acetylcholine followed by the activation of postsynaptic receptors, of which two types of acetylcholine receptors have been identified: muscarinic and nicotinic. The muscarinic receptor family is distributed throughout the body, having an important role in diseases outside the nervous system—specifically, the cardiovascular system. In humans and animals, five subtypes of muscarinic receptors (M1–M5) have been identified; M1 and M3 are involved in acetylcholine-induced vasodilation and are expressed in both the vascular endothelium and smooth muscle [139]. Several studies have shown the importance of M3 associated to its regulation, signal transduction and its distribution (in heart and blood vessels) to regulate endothelial NO release. Therefore, M3 is an attractive therapeutic target in hypertension and cardiovascular diseases [140].

At the vascular level, the mechanisms by which the metabolic syndrome and diabetes increase RNS, and the mechanisms by which RNS modify the vessels, are not yet well-known. Increased extracellular high glucose concentrations induce a deregulation of the reactive oxygen and nitrogen-generating pathways. In this sense, high glucose alters the balance of prostacyclin I_2_/thromboxane A_2_ by the 3-tyrosine nitration of the prostacyclin synthase enzyme via ONOO^−^, decreasing the synthesis of prostacyclin I_2_ by the endothelial cells [141]. Prostacyclin I_2_ relaxes isolated vascular strips and is an endogenous inhibitor for platelet and leukocyte activation. Decreased prostacyclin I_2_ has been linked to platelet hyperagregability, increased adhesiveness and increased release of the prostaglandin H_2_ synthase/thromboxane A_2_ in diabetic patients. Prostacyclin I_2_ not only prevents platelet adhesion, but it also disrupts existing aggregates in vitro, and it also acts on circulating aggegates. Nitration thus inhibits the vasodilators, growth-inhibiting, antithrombotic and antiadhesive effects of prostacyclin I_2_ and increases the release of the potent vasoconstrictor, prothrombotic, growth- and adhesion-promoting agents, prostaglandin H_2_ synthase and thromboxane A_2_. Therefore, at the vascular level, NSS, through ONOO^−^-induced 3-tyrosine nitration, damages the eNOS and COX-1. This impairs vasodilation by diminishing the capacity of vessels to synthesize the vasodilators NO and prostacyclin I_2_ [142].

Hypertension involves not only peripheral factors and regulation of vascular function but, also, complex signaling mechanisms throughout the central nervous system. ROS and NOS are thus involved in blood pressure deregulation in the central nervous system. They serve as signaling molecules within neurons of cardiovascular regulatory centers and in the regulation of the sympathetic nervous system [143]. The sympathetic system regulates vessel tone and heart functioning. Central regulation also participates in volume homeostasis, determining the water and salt balance in the kidneys and modulating the circulatory release of blood pressure-regulating hormones. Alterations in the central nervous system and endoplasmic reticulum environment by NSS lead to long-term changes in neural function and regulate the synthesis and release of molecules involved in the central nervous system-driven hypertension produced by the metabolic syndrome. These molecules include Ca^2+^, ROS and transcription factors such as NF-κB and AP-1. Interestingly, cellular perturbations known to cause endoplasmic reticulum stress, including oxidative and NSS, and alterations in intracellular Ca^2+^ are prevalent in hypertension [144].

Evidence indicates that endoplasmic reticulum stress, which is caused in part by NSS, plays an important pathophysiological role in metabolic syndrome-related hypertension; the endoplasmic reticulum is responsible for protein synthesis, folding and trafficking and is also recognized as a primary sensor of cellular stress. Any conditions that challenge the normal endoplasmic reticulum function, such as increased protein synthesis, alterations in the cellular redox status or disturbances in the intracellular Ca^2+^ levels caused in part by the metabolic syndrome, may induce endoplasmic reticulum stress and lead to the activation of intracellular signaling cascades known as the unfolded protein response. This may facilitate the adaptation to acute cellular alterations and resume the endoplasmic reticulum homeostasis [145]. The participation of NSS and its effects in cardiometabolic diseases are summarized in Table 1.

## 10. Conclusions

NSS represents a redox imbalance and contributes to it. It is a highly oxidizing condition that participates in the deterioration of organs and systems. It is currently being found to participate in many cardiometabolic pathologies such as hypertension, diabetes, endothelial dysfunction and cardiovascular diseases, among others, and its mechanisms of action are still being elucidated. This review summarizes much of the information that is nowadays reported. The RNS are stable metabolites of the NO pathway, which react three times faster and have a longer mean half-life than O_2_-. Therefore, they may cause irreversible damage to cell membranes, proteins, enzyme activity, the endoplasmic reticulum and nucleic acids, leading to necrosis and cellular death. The information exposed in this review might be useful for the future proposal of anti-NSS therapies. However, there remain future challenges, which include the validation of available biomarkers of NSS damage in animal and human studies, the accurate determination of the causal relation with diseases and the response to anti-NSS intervention. It would also be necessary to assess the long-term effects of nitrosative damage by well-designed, randomized and controlled trials in humans.

## Figures and Tables

**Figure 1 molecules-25-02555-f001:**
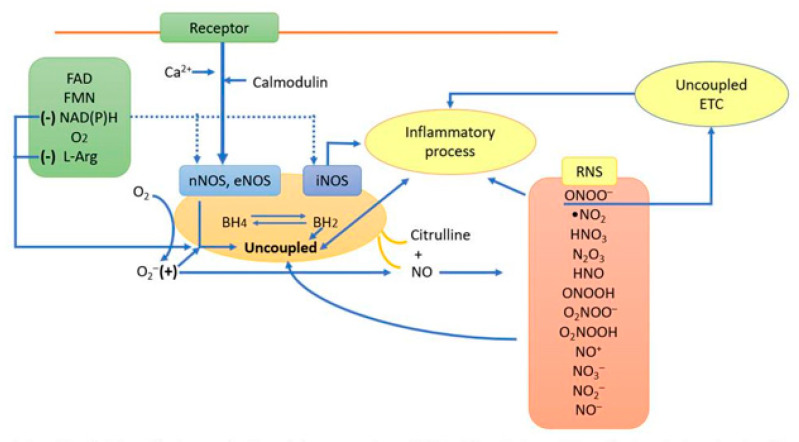
Interrelationship between the synthesis of nitric oxide and the generation of RNS. The RNS are formed in the presence of high O_2_ concentrations—and when there is an abnormal increase in the level of NO produced by the iNOS and/or by the uncoupled eNOS and nNOS. Abbreviations: FAD = flavin adenine dinucleotide, FMN = flavin mononucleotide, NAD(P)H = nicotinamide adenine dinucleotide, O_2_ = molecular oxygen, RNS = reactive nitrogen species, NO = nitric oxide, ONOO^−^ = peroxynitrate, •NO_2_ = nitrogen dioxide, HNO_3_ = peroxynitrous acid, N_2_O_3_ = dinitrogen trioxide, HNO = nitroxyl, ONOOH = peroxynitrous acid, O_2_NOO^−^ = peroxynitrate, O_2_NOOH = peroxynitric acid, NO^+^ = nitrosonium cation, NO_3_^−^ = nitrate, NO_2_^−^ = nitrite, NO^−^ = nitroxyl anion, RNS = reactive nitrogen species, ETC = electron transport chain, BH4 = tetrahydrobiopterin, BH2 = dihydrobiopterin and L-arg = l-arginine. nNOS, eNOS and iNOS = neuronal, endothelial and inducible nitric oxide synthase.

**Figure 2 molecules-25-02555-f002:**
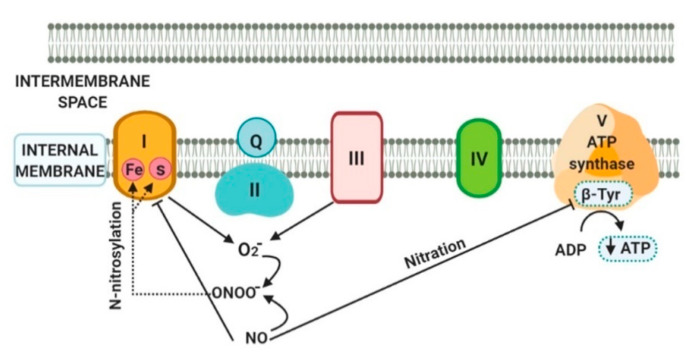
Interaction between nitrosative stress (NSS) and the mitochondrial proteins. The nitration of Tyr residues in the catalytic β subunit of the complex V decrease adenosine triphosphate (ATP) levels. The ONOO^−^ impairs complex I by S-nitrosylation affecting the iron-sulphur centers.

**Figure 3 molecules-25-02555-f003:**
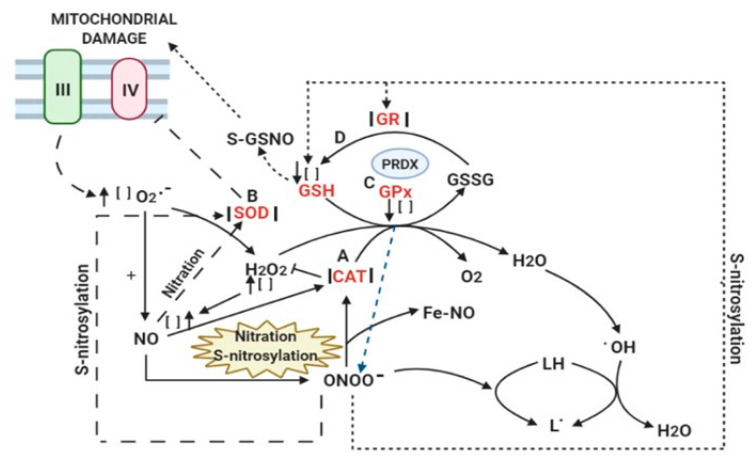
Nitrosative stress and the antioxidant system. A deficient of the antioxidant defense mechanisms favor the processes of nitration and nitrosylation of different enzymes. Reaction (**A**): Nitration and S-nitrosylation influence the activity of the CAT that affects its binding to H_2_O_2_. Reaction (**B**): The S-nitrosylation of the catalytic site in the SOD may lead to the loss of its activity, and, also, a high NO concentration enhances O_2_^−^ production by inhibition of the cytochrome oxidase. Reaction (**C**): The lack of GPX 1 enhances ONOO^−^ survival. (**D**): There is an inhibition of GR activity and a reduced concentration of GSH after the S-nitrosylation; the reaction with GSH generates S-GSNO. Abbreviations: SOD = superoxide dismutase, CAT = catalase, GPx = glutathione peroxidase, PRDX = peroxiredoxin, GR = glutathione reductase, GSH = glutathione, GSSG = oxidized glutathione, S-GSNO = S-nitroso glutathione, Fe-NO = ferric-nitrosyl, NO = nitric oxide, ONOO^−^ = peroxynitrite, OH^−^ = hydroxyl radical, LH = polyunsaturated fatty acid, L = carbon-center lipid radical and [] = concentration.

**Figure 4 molecules-25-02555-f004:**
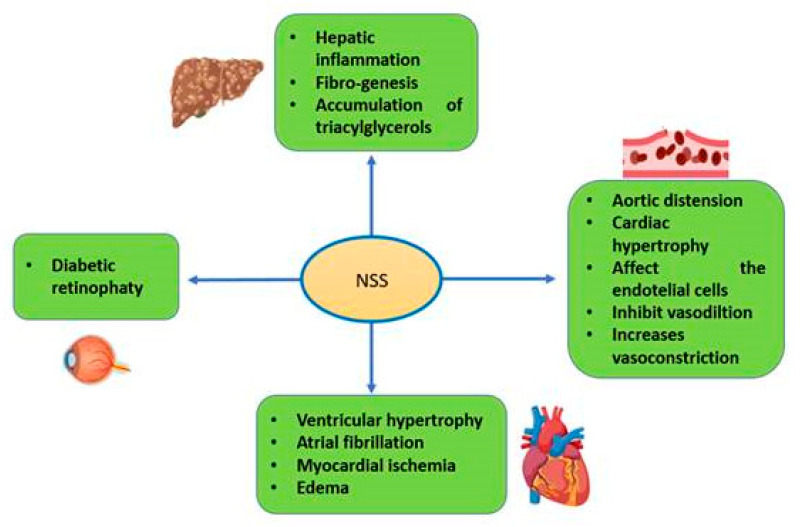
Relation of NSS with some metabolic disorders. Abbreviations: NSS = nitrosative stress.

**Table 1 molecules-25-02555-t001:** Participation of nitrosative stress in cardiometabolic complications.

Metabolic Disorders	Alterations
Diabetes mellitus	↑ONOO^−^, 3-nitrotyrosine, iNOS, NAD(P)H oxidase, O_2_^−^ (uncoupled eNOS), HbA1c, ICAM, VCAM [81,88,89,90]
Diabetic retinopathy	↑Aldose reductase activity [97] ↓GSH, Ascorbate, Taurine, Na/KATP-ase activity [114] ○PARP-1 [97] ●HO-1 [101]
Obesity	↑iNOS, NO [101]
Fatty liver	↓ApoB100, Acyl-coenzyme-A expressions [105]
Cardiovascular disorders	↑3-nitrotyrosine, ONOO-, iNOS [107] ↓Intracellular Ca2+, Ca2p, ATP, PARP-1 [108]
Endothelial dysfunction	↑O2^−^, ONOO^−^, iNOS, Interleukins, VCAM, ICAM, P-seletins, PKC, NFKB [14,129] ●SOD [61]
Hypertension	↑ONOO^−^, COX-2 activity, NAD(P)H expression [20,128,137] ↓L-arg, NO, PGI2 [128,129,141].
○Activation ●Inactivation ↑Increase ↓Decrease	

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
