# Peer review of "Nitrosative Stress and Its Association with Cardiometabolic Disorders"

_molecules, 2020, doi:10.3390/molecules25112555_

Round 1

Reviewer 1 Report

In this manuscript the authors provide a thorough review regarding the role of Reactive nitrogen species (RNS) in cardiometabolic disorders. Overall, the authors have done a substantial amount of literature review regarding this theme; however the English language needs great improvement. It is extremely difficult to read and follow the content of this review article. In my opinion the manuscript has to be thoroughly edited prior to being considered for publication. Overall, I believe that the content of this review article is interesting and of use to researchers within the area, particularly focused on cardiovascular diseases. I think that this manuscript is appropriate for publication in Molecules journal. However it needs major revisions regarding the English language prior to consideration for publication.

Example:

Abstract:

“Currently nitrosative stress (NSS) is being implied in the pathogenesis of many diseases but its mechanisms of damage production are still being elucidated.”

In this short sentence one can find bad construction/structure of the sentence, and substantial (critical) grammatical errors.

The link between the different sections should also be improved. This would provide a better flow and structure to the manuscript.

Author Response

May 28, 2020

Dear

Ms. Milena Mirkovic, B.Sc.

Assistant Editor, MDPI AG

Thanks you for your letter and we fully appreciate the time spent by the reviewers in evaluating our paper as well as their comments and advice. We added a response letter to questions of the third round for each reviewer separately, and I send the new version of the review paper with title NITROSATIVE STRESS AND ITS ASSOCIATION WITH PATHOLOGICAL DISORDERS. Manuscript ID: molecules-813544, By the authors: Israel Pérez-Torres, Linaloe Manzano Pech, María Esther Rubio-Ruíz, María Elena Soto and Verónica Guarner-Lans. We have tried to follow their suggestions and enclose our replies, which are marked in red.        Best regards   Israel Pérez-Torres PhD

Referee 1

Comments and Suggestions for Authors

In this manuscript the authors provide a thorough review regarding the role of Reactive nitrogen species (RNS) in cardiometabolic disorders. Overall, the authors have done a substantial amount of literature review regarding this theme; however the English language needs great improvement. It is extremely difficult to read and follow the content of this review article. In my opinion the manuscript has to be thoroughly edited prior to being considered for publication. Overall, I believe that the content of this review article is interesting and of use to researchers within the area, particularly focused on cardiovascular diseases. I think that this manuscript is appropriate for publication in Molecules journal. However it needs major revisions regarding the English language prior to consideration for publication.

Example:

Abstract:

Question 1

“Currently nitrosative stress (NSS) is being implied in the pathogenesis of many diseases but its mechanisms of damage production are still being elucidated.”

In this short sentence one can find bad construction/structure of the sentence, and substantial (critical) grammatical errors.

Answer

Done, the manuscript was reviewed in by native English speaker, and was modified in something any sections, the changes are marked in red.

Question 2

The link between the different sections should also be improved. This would provide a better flow and structure to the manuscript.

Answer

Done, the link between the different sections was improved. Thanks you for your comments, and for the time spent reviewing the manuscript.

Referee 2

Question 1

English language and style are fine/minor spell check required

Answer

Done, the manuscript was reviewed in by native English speaker, and was modified in something any sections, the changes are marked in red.

Question 2

I find this version of the manuscript acceptable.

Answer

Thanks you for your comments, and for the time spent reviewing the manuscript

Reviewer 2 Report

I find this version of the manuscript acceptable.

Author Response

(The authors gave the same response as above.)

Round 2

Reviewer 1 Report

The authors have answered all comments satisfactorily and for this reason I feel that the manuscript is suitable for publication.

Best regards

This manuscript is a resubmission of an earlier submission. The following is a list of the peer review reports and author responses from that submission.

Round 1

Reviewer 1 Report

General comments:
In general, I did not find strong motivation to publish this long narrative review. It seems that the authors just wrote whatever they know about RNS without clear focus. There are many reviews on RNA, and the authors are encouraged to cite them around L44, stating that how this review is different from these existing reviews (e.g., update or special focus). The logical flow is often not clear, and specific terms are sometimes not used correctly. I tried to point out some of them below, but my attempt is not comprehensive. There are some grammatical errors to be corrected, but the content needs to be revised before considering the English problem.

Meanwhile, after section 10 of this review seems relatively better and interesting to me, although the description is still superficial. I would consider that it is difficult to fit general RNS topics into a single review (like this review with 344 references); it will be a textbook. RNS and diseases, although it may be still overwhelming, can be a single review if it has a special point of view. RNS and a single disease will be a review of appropriate focus. I would suggest to carefully select the topic to include in the revision.

Specific comments:
L15 and others: Nitric oxide is sometimes abbreviated as NO, but sometimes ON. It is discouraging that one of the critical keywords of the review has this kind of inconsistency.
L20: This sentence is not clear enough to me, especially the word "oxygen free radicals". Probably the authors meant RNS?
L39: Same as above. NRS should be RNS.
L43: "It is also present" sounds not appropriate here. As the authors stated in L28-29, RNS is are generated by many normal physiological processes and always present.

L60: Remove comma after [9].
L63: "...binds to two molecules of molecular O2 and 1.5 moles of reducing equivalent" Is this true? One mole contains 6.022 x 10^23 molecules. It seems that the authors are describing the molar ratio of the reaction, but mixed up it with moles, which is basic.

L81: Please specify "The reaction of this RNS". Reaction with what?
L92: "Therefore" in this line probably refers the fact that maximum ONOO- concentration is associated with maximum protein tyrosine nitration. Accordingly the sentence citing Ref. 18 is misplaced.

L168: "One of the main products..." but the authors mention three molecules here. Also, a conjugated diene is a set of two double bonds separated by a single bond, and not a name of a molecule.

L177: Please clarify the meaning of "that which addresses the review".
L181: Please clarify the meaning of this sentence.
L184: Cysteine is the second most abundant... Can you add a citation for this?

L204: This section has three major topics, general mitochondria (L205 - 216), mitochondria in the liver and some hepatic disease (L216-L222), and mitochondria in other organs (L223 - L288). To me the current paragraph structure is not appropriate.

L210: "Proteins" in this sentence means mitochondrial proteins?
L245: This sentence contains three relative pronouns (that, that, which) and is not kind to readers. Also, proteins and genes are mixed up in this sentence. In the first place, I wonder whether this paragraph is necessary. The most part of this paragraph is about PARP-1. This is related to RNS but not RNS itself.

L274: Spell out CAT here, or show the abbreviation in L212.
L352: If the aim of this review is the pathological disorders, the association between NSS and zinc is out of scope.
L407: Spell out BP

Author Response

April 2020

Editorial Office

MDPI molecules

We fully appreciate the time spent by the reviewers in evaluating our paper as well as their comments and advice. I send, the new version of the review paper "NITROSATIVE STRESS AND ITS ASSOCIATION WITH PATHOLOGICAL DISORDERS". By the authors: Israel Pérez-Torres, Linaloe Manzano Pech, María Esther Rubio-Ruíz, María Elena Soto and Verónica Guarner-Lans. We have tried to follow their suggestions and enclose our replies, which are marked in red.        Best regards   Israel Pérez-Torres PhD

Referee 1

General comments:
In general, I did not find strong motivation to publish this long narrative review. It seems that the authors just wrote whatever they know about RNS without clear focus. There are many reviews on RNA, and the authors are encouraged to cite them around L44, stating that how this review is different from these existing reviews (e.g., update or special focus). The logical flow is often not clear, and specific terms are sometimes not used correctly. I tried to point out some of them below, but my attempt is not comprehensive. There are some grammatical errors to be corrected, but the content needs to be revised before considering the English problem.

Meanwhile, after section 10 of this review seems relatively better and interesting to me, although the description is still superficial. I would consider that it is difficult to fit general RNS topics into a single review (like this review with 344 references); it will be a textbook. RNS and diseases, although it may be still overwhelming, can be a single review if it has a special point of view. RNS and a single disease will be a review of appropriate focus. I would suggest to carefully select the topic to include in the revision.

Specific comments:
Answer: L15 and others: Nitric oxide is sometimes abbreviated as NO, but sometimes ON. It is discouraging that one of the critical keywords of the review has this kind of inconsistency.
Response 1: A apologize, by the type graphics error, now has been change the abbreviation

Answer: L20: This sentence is not clear enough to me, especially the word "oxygen free radicals". Probably the authors meant RNS?
Response: If, oxygen free radicals was change by RNS

Answer: L39: Same as above. NRS should be RNS.

Response 2: A apologize, by the type graphics error, now has been change the abbreviation

Answer: L43: "It is also present" sounds not appropriate here. As the authors stated in L28-29, RNS is are generated by many normal physiological processes and always present.

Response 3: This was changed

Answer: L60: Remove comma after [9].

Response 4: the comma was removed

Answer: L63: "...binds to two molecules of molecular O2 and 1.5 moles of reducing equivalent" Is this true? One mole contains 6.022 x 10^23 molecules. It seems that the authors are describing the molar ratio of the reaction, but mixed up it with moles, which is basic.

Response 5: it is correct; the phrase was changed (in red)

Answer: L81: Please specify "The reaction of this RNS". Reaction with what?

Response 6: The phrase the reaction of this RNS, was completed
Answer: L92: "Therefore" in this line probably refers the fact that maximum ONOO- concentration is associated with maximum protein tyrosine nitration. Accordingly the sentence citing Ref. 18 is misplaced.

Response 7: The reference was moved of the phrase

Answer: L168: "One of the main products..." but the authors mention three molecules here. Also, a conjugated diene is a set of two double bonds separated by a single bond, and not a name of a molecule.

Response 8: the phrase one of the main products, was changed by: Between products of lipid peroxidation are.

Answer: L177: Please clarify the meaning of "that which addresses the review". And L181: Please clarify the meaning of this sentence.
Response 9: The phrase was changed by: The Figure 2 describes the relationship between NSS and metabolic disorders which are addressed in this review.

Answer: L184: Cysteine is the second most abundant... Can you add a citation for this?

Response 10: The reference was added

Answer: L204: This section has three major topics, general mitochondria (L205 - 216), mitochondria in the liver and some hepatic disease (L216-L222), and mitochondria in other organs (L223 - L288). To me the current paragraph structure is not appropriate.

Response 11: Thanks you for your comment, was eliminated the words liver, hepatic disease and others organs, now only this section speak of the mitochondria in general

Answer: L210: "Proteins" in this sentence means mitochondrial proteins?

Response 12: If, it is correct, now in the 216 line is mostrated the phrase mitochondrial proteins

Answer: L245: This sentence contains three relative pronouns (that, that, which) and is not kind to readers. Also, proteins and genes are mixed up in this sentence. In the first place, I wonder whether this paragraph is necessary. The most part of this paragraph is about PARP-1. This is related to RNS but not RNS itself.

Response 13: Thanks you for your comment, The words was changed by other synonyms

Answer: L274: Spell out CAT here, or show the abbreviation in L212.

Response 14: The abbreviation by CAT was added

Answer: L352: If the aim of this review is the pathological disorders, the association between NSS and zinc is out of scope.

Response 15: The zinc metalloid in very important in this pathological process, so we think it's important to include it

Answer: L407: Spell out BP

Response 16: The abbreviation by BP was added

Referee 2

In this manuscript the authors perform a review on the alterations caused by reactive nitrogen species (RNS) in the cellular redox system, subsequently causing nitrosative stress. The authors also discuss the participation of NRS in several diseases, such as hypertension, diabetes, endothelial dysfunction, cardiovascular diseases, erectile dysfunction, cancer, renal failure, inflammation, atherosclerosis, and aging. The authors perform a very thorough examination of the current literature, having referenced 344 papers. The subject of the review is of interest to a wide variety of researchers, working in different research fields. Overall, the manuscript needs comprehensive editing of the English language throughout, with a particular focus for the use of singular and plural, and the use of prepositions. There are also issues regarding formatting, such as different font sizes. I believe that this manuscript is appropriate for publication in the Molecules journal upon minor revisions as detailed bellow:

Response 1, Thanks you for your comments. The editing of the English language was made, also of the formatting edition

General comments:

Answer: Some protein abbreviations are used without introduction of the complete name of the protein. This should be checked throughout the manuscript.

Response 2: the abbreviation of many proteins was eliminated and now are mentioned without yourself abbreviation.

Answer: Excessive use of abbreviations, some of which are only used once or twice in the manuscript. The authors should review the use of abbreviations and restrict them to words or small group of words that are used frequently in the manuscript.

Response 3: the abbreviation of many proteins was eliminated and now are mentioned without yourself abbreviation.

Specific comments:

Answer: Line 45: “2.Interrelationship of NO, its synthesis and the generation of RNS” – the term interrelationship relates to the relationship between two or more molecules – for this reason the title as it is does not make sense. Interrelationship of NO with …

Response 4, Thanks you for your comment. It is was changed, according to your suggestion

Answer: Line 49: Where it reads: “These concentrations also participate in host cell defenses”, it should read: NO at these concentrations also participate in host cell defense mechanisms.

Response 5, Thanks you for your comment. It is was changed, according to your suggestion

Answer: Line 59: Where it reads: “the reaction requires of molecular O2”, it should read: the reaction requires molecular O2.

Response 6, Thanks you for your comment. It is was changed, according to your suggestion

Answer: Line 70: Figure legend is incomplete. It has only a title and abbreviations, but no explanation/summary of the diagram/Figure content.

Response 7: It is added, now the figure legend in more complete

Answer: Lines 80-81: Where it reads: “NO, in the presence of O2– can form all of the RNS molecules, but the predominant residue formed is ONOO–“, it should read: NO, in the presence of O2– can form all of the RNS, but the predominant molecule formed is ONOO–, since ONOO– is not a residue.

Response 8, Thanks you for your comment. It is was changed, according to your suggestion

Answer: Line 85: Where it reads: “ONOO– rapidly decays to its protonated from”, it should read: ONOO– rapidly decays to its protonated form.

Response 9, Thanks you for your comment. It is was changed, according to your suggestion

Answer: Line 109: “in heme peroxidase enzymes, such as myeloperoxidase”. The authors should add more examples.

Response 10: Done, now was added more examples of the peroxidase enzymes

Answer: Lines 121-122: Where it reads: “causing a positive feedback process in the uncoupled enzymes, contributing to NSS”, it should read: causing a positive feedback loop in the uncoupled enzymes, contributing to NSS formation.

Response 11, Thanks you for your comment. It is was changed, according to your suggestion

Answer: Lines 134-137: Where it reads: “However, this does not prevent uncoupling of eNOS and is associated with ventricular remodeling, which suggest that the presence and participation of uncoupled eNOS in cardiomyocytes and/or fibroblasts in cardiac hypertrophy and can be considered as the missing link between atrial fibrillation and chronic non-ischemic cardiomyopathy”, it should read: However, this does not prevent uncoupling of eNOS and is associated with ventricular remodeling, which suggest the presence and participation of uncoupled eNOS in cardiomyocytes and/or fibroblasts in cardiac hypertrophy which can be considered as the missing link between atrial fibrillation and chronic non-ischemic cardiomyopathy.

Response 12, Thanks you for your comment. It is was changed, according to your suggestion

Answer: Lines 176-178: Where it reads:” The Figure 2 describes the relation of the NSS with some metabolic disorders that which addresses the review”, it should read: Figure 2 describes the relationship between NSS and metabolic disorders which will be addressed in this review.

Response 13, It is was changed, according to your suggestion

Answer: Lines 185-186: Where it reads: “However, only a small percentage of cysteine is susceptible to modification by ONOO– [56].”, it should read: However, only a small percentage of cysteine residues within proteins are susceptible to modification by ONOO–. The authors should explain what characteristics make a cysteine residue susceptible to ONOO- oxidation (reactive cysteine).

Response 14, done it is was changed, according to your suggestion

Answer: Lines 206-207: The authors should develop/explain the following, for instance by giving specific examples: “The mitochondria play a critical role as antioxidant defences”.

Response 15: Done, it is added in the 209-212 lines

Answer: Line 224: Where it reads: “Tyr nitration is the major mechanisms responsible for the irreversible inhibition of complex I provoked by adding exogenous NO that increase ONOO– levels”, it should read: Tyr nitration is the major mechanism responsible for the irreversible inhibition of mitochondrial complex I. This occurs via addition of exogenous NO leading to increases in ONOO– levels. The word: “provoked” is not appropriate in this context.

Response 16, done it is was changed, according to your suggestion

Answer: Line 261: Where it reads: “The PARP pharmacological inhibition or genetic deletion of PARP-1 preserves cellular NAD+”, it should read: PARP-1 pharmacological inhibition or genetic deletion preserves cellular NAD+.

Response 17, done it is was changed, according to your suggestion

Answer: Lines 262-263: “ATP pools in oxidatively and/or nitrosatively stressed endothelial cells (as well as many other cell types), thereby allowing them to function normally.” It would be more appropriate: ATP pools in oxidatively and/or nitrosatively stressed cells, thereby … References are missing at the end of this sentence.

Response 18, Thanks you for comment. Done it is was changed, according to your suggestion

Answer: Line 288: Where it reads: “On another hand”, it should be: On the other hand. However I do not think that this is appropriate in the context of the text, and in fact should be removed.

Response 19: Done It was changed

Answer: Line 291: “This last isoform has a molecular mass of 30kDa”. I do not understand why it is relevant to put the molecular weight of Cu/Zn-SOD. I suggest removing this information.

Response 20: This phrase was eliminated

Answer: Line 312: Where it reads: “The glutathione peroxidase (GPx) isoform family”, it should read: The glutathione peroxidase (GPx) family.

Response 21: Done it is was changed, according to your suggestion

Answer: Lines 313-314: Where it reads: “The GPx1 is the main antioxidant enzyme preventing the accumulation of damaging intracellular by H2O2, it uses GSH and belongs to the GPxs family”, it should read: The GPx1 is the main antioxidant enzyme preventing the accumulation of intracellular damage by H2O2, in a GSH dependent manner.

Response 22: This phrase was eliminated

Answer: “and belongs to the GPxs family” – this should be removed as it is already mentioned at the beginning of the paragraph.

Response 23: This phrase was eliminated

Answer: Lines 319-321: “In the case of 2-Cys peroxiredoxin, small oxide-reductases with redox centers comprising two Cys-residues regenerate the catalytic site, whereas a cysteine residue in GPx is reduced by GSH [103].” This sentence needs thorough editing of the English language for clarification of its content.

Response 24: the sentence was edited

Answer: Line 365: Where it reads: “that acts in the control the Zn2+ concentration”, it should read: “that acts in the control of Zn2+ concentration(s)”.

Response 25: Done it is was changed, according to your suggestion

Answer: Line 372: Where it reads: “resulting in a vicious circle”, it should be: resulting in a vicious cycle.

Response 26: Done it is was changed, according to your suggestion

Answer: Lines 382-386: The following sentence needs thorough editing of the English language: “Several reports have concluded that Zn2+ functions as a cyto-protective agent, defending cells

383 against oxidative insults and stimuli that induce apoptosis. This protective effect includes serving as a factor in the maintenance and reparation of the cellular membranes activation of anti-apoptotic signal transduction [128], inhibition pro-apoptotic of the NFĸB, activator protein 1 (AP-1) [129], antagonism of LPO and inhibition of caspases and endonucleases [130].”

Response 27: the paragraph was edited

Answer: Line 435: Where it reads: “neovascularization and macular edema and can also occur”, it should read: neovascularization and macular edema can also occur.

Response 28: Done it is was changed, according to your suggestion

Answer: Lines 459-463: “Eales disease is an idiopathic inflammatory venous occlusion condition with male predominance (99%) that affects the peripheral retina of young adults (15 to 45 years). Inflammation is the main clinical feature of this pathology [159], where there is also an increases the aldose reductase activity, a decreases of the Na/KATP-ase activity [160] and an imbalance in the NAD+/NADH and NADP+/NAD(P)H redox levels. These changes are associated with NSS [161].” The authors should explain in more detail how NSS induces/is associated with the changes mentioned above.

Response 29: Done, the paragraph was edited and now between 482-488 was mentioned Inflammation is the main clinical feature of this pathology [163] and this is associate with increases the aldose reductase activity which catalyze NAD(P)H dependent reduction of glucose to sorbitol [158]. The increase of the sorbitol concentrations leads to decreases of the Na/KATP-ase activity and an imbalance in the NAD+/NADH and NADP+/NAD(P)H redox leves [162, 164]. These changes are associated with NSS through formation of advanced glycation end products, activation of PKC and mitogen activated protein kinase [162, 165], and an increased aldose reductase activity which is linked to destruction of capillary retinal cells [16].

Answer: Line 492: Where it reads: “NSS plays an important role in the pathogenesis of CVD”, it should read: NSS plays an important role in the pathogenesis of cardiovascular diseases (CVD).

Response 30: Done it is was changed, according to your suggestion

Answer: Lines 558-560: The following sentence needs editing of the English language: “The regulatory roles of NO and ONOO– by endothelial cells are controlled by their production rates and their concentrations and effect on vascular smooth muscle cells (VSMC) [199].”

Response 31: the paragraph was edited

Answer: Line 564: “SOD inactivation also increases NSS and induces tissue damage [202].” The authors should develop this theme and give some examples.

Response 32: In these lines 299-315, sections: antioxidant enzymes, this was developed

Answer: Lines 579-581: The following sentence needs editing of the English language: “The main NO receptor initiating the canonical signaling pathway is the sGC receptor which is responsible for the enzymatic e conversion of GTP into cGMP, and is crucial for the modulation of the tone vascular.”

Response 33: This section was edited

Answer: Line 604: The English language in the section: “11.2. NSS and essential arterial hypertension and metabolic syndrome-related hypertension” is particularly problematic (especially in the first 4-5 paragraphs) and should be thoroughly examined. I would suggest shorter sentences in this section.

Response 34: This section was edited

Answer: Lines 748-749: Where it reads: “ADMA can be transported into the cells through the inducible L-arg transporter CAT-2B, that is the same transporter that uses the L-arg.”, it should read: ADMA can be transported into the cells through the inducible L-arg transporter CAT-2B, that is the same transporter for L-arg. Alternatively, the last part of the sentence “that is the same transporter for L-arg” should be removed, because it is redundant.

Response 35: Done it is was changed, according to your suggestion

Lines 754-755: “and result in a hypoxic response” – the authors should explain what is hypoxia?

Response 36: Done in the line 786 this was added

Answer: Line 756: “Activated HIF-1α decreases mitochondrial biogenesis by the HIF-1α–PGC-1α–TFAM axis” – what is PGC and TFAM? The authors use only the abbreviations for these proteins.

Response 37: Done, the abbreviations was changed by the name of the enzymes

Answer: Lines 762-763: The following sentence is confusing: “Moreover, the exogenous administration of inhaled NO can alter the endogenous pulmonary endothelial function [278].” Do the authors mean: The inhalation of NO can alter … please clarify.

Response 38: This sentence was edited

Answer: Lines 762-772: This paragraph needs revision of the English language.

Response 39: This paragraph was edited

Answer: Line 781: “During severe sepsis” – the authors should add the definition of sepsis.

Response 40: Done in the lines, 814-815 this was added

Answer: Line 803: In the section “13. NSS and cancer”, the authors show a focus on prostate cancer and very little detail regarding other types of cancer. This appears unbalanced. Is there more studies focus on NSS in prostate cancer? Otherwise the authors should develop this section including studies focused in other types of cancer with more detail.

Response 41: thanks you for your comment. However, if we developed this section further, the document would be longer and the other referees commented that will shorten, so we only modified the subtopic to emphasize in these examples.

Answer: Suggestion: Figure with diagrams for the different NOS.

Response 42: Thanks you for your suggestion. However, we believe that figure 1 is sufficient because it groups together the three NOS isoforms

Referee 3

This is a comprehensive, detailed and multi-leveled coverage of the different biopharmaceutical implications of nitric oxide stress in diverse diseases, tissues and processes. The topic is of considerable importance, which justifies the assembly of new information into a coherent entity which can be of use to many and is a strength point of this review.  However, this topic had already been covered by many for a number of years, such that publishing a new review on this issue calls for a new point of view, a fresh concept or a link between the impact of nitric oxide stress in different tissues and organs. Unfortunately, I have failed to find such a fresh line of thought or any novel link between the different sub-topics covered in this lengthy text, which is a major point of weakness in this manuscript and calls for a substantial revision, as is listed below.

  1. Most importantly, no critical thinking is seen throughout this long text, which simply lists all of those instances where NO stress may exert some damage, without comparing the severity of the listed phenomena or explaining the order in which those are presented.

Response 1: Thanks you for your comment. However, nitrogen species are very important in the redox imbalance, and this review focuses in the alterations that cause currently nitrosative stress is being implied in the pathogenesis of many diseases but its mechanisms of damage production are still being elucidated. Why that in paper we summarize current knowledge on the participation of nitrosative stress in the pathology of these diseases and its possible mechanisms of action

  1. In particular, body-brain communication processes and their role as transmitting the message of impaired NO pathways are missing. Specifically, the links between NO imbalance and muscarinic regulation in arteries should be covered.

Response 2: Done, in the 570-574, and 649-660 lines it was added

  1. Further, the great majority of the human genome is devoted to long and short non-coding transcripts which nevertheless have key biological roles, but this review totally ignores that aspect of human biology which is badly absent here. Examples include microRNA regulators of inflammation such as miR-132 and its impact on NO stress.

Response 3: Done, in the 368-377 lines it was added.

  1. Also, the text is far too long for the ordinary scientist to go through. Try to cut the length of this text by half, which further will enforce focusing on the more important issues.
  2. Response: Thanks you for your comment. However, the topic is very broad and complicated, and we believe that the topics covered by the review are of interest, so we do not consider shortening it.
  3. Adding a table or two where the less central phenomena can be listed may be of assistance in this task of shortening this lengthy review.
  4. Response: Thanks you for your suggestion. The table was added
  5. Add clear and informative 3-5 illustrations highlighting the main messages, in the form of a graphical abstracts which will open each section. Those can be of major importance and will also aid in selecting the topics of interest in this long text.
  6. Response: Thanks you for your suggestion. Was added 2 figures more.
  7. Additionally, the bibliography is excessively long and it appears as if the citations were not filtered at all, such that one finds numerous citations of negligible journals and lacks the capacity to appreciate which of the citations is more important or worthy. To correct this flaw, the bibliography should be sharply cut, with those journals of higher impact factors preferred as they may be considered to have been subjected to critical reviewing by professional experts; and newer references might be preferred considering that such reviews might be more-timely than its present version.
  1. Response: The information that was cited in the review, is what the mentions the reference on the theme, and we believe that be important for the researchers that worked in the theme, independent of the impact of the magazine in which it was published.

Reviewer 2 Report

In this manuscript the authors perform a review on the alterations caused by reactive nitrogen species (RNS) in the cellular redox system, subsequently causing nitrosative stress. The authors also discuss the participation of NRS in several diseases, such as hypertension, diabetes, endothelial dysfunction, cardiovascular diseases, erectile dysfunction, cancer, renal failure, inflammation, atherosclerosis, and aging. The authors perform a very thorough examination of the current literature, having referenced 344 papers. The subject of the review is of interest to a wide variety of researchers, working in different research fields. Overall, the manuscript needs comprehensive editing of the English language throughout, with a particular focus for the use of singular and plural, and the use of prepositions. There are also issues regarding formatting, such as different font sizes. I believe that this manuscript is appropriate for publication in the Molecules journal upon minor revisions as detailed bellow:

General comments:

Some protein abbreviations are used without introduction of the complete name of the protein. This should be checked throughout the manuscript.

Excessive use of abbreviations, some of which are only used once or twice in the manuscript. The authors should review the use of abbreviations and restrict them to words or small group of words that are used frequently in the manuscript.

Specific comments:

Line 45: “2.Interrelationship of NO, its synthesis and the generation of RNS” – the term interrelationship relates to the relationship between two or more molecules – for this reason the title as it is does not make sense. Interrelationship of NO with …

Line 49: Where it reads: “These concentrations also participate in host cell defenses”, it should read: NO at these concentrations also participate in host cell defense mechanisms.

Line 59: Where it reads: “the reaction requires of molecular O2”, it should read: the reaction requires molecular O2.

Line 70: Figure legend is incomplete. It has only a title and abbreviations, but no explanation/summary of the diagram/Figure content.

Lines 80-81: Where it reads: “NO, in the presence of O2– can form all of the RNS molecules, but the predominant residue formed is ONOO–“, it should read: NO, in the presence of O2– can form all of the RNS, but the predominant molecule formed is ONOO–, since ONOO– is not a residue.

Line 85: Where it reads: “ONOO– rapidly decays to its protonated from”, it should read: ONOO– rapidly decays to its protonated form.

Line 109: “in heme peroxidase enzymes, such as myeloperoxidase”. The authors should add more examples.

Lines 121-122: Where it reads: “causing a positive feedback process in the uncoupled enzymes, contributing to NSS”, it should read: causing a positive feedback loop in the uncoupled enzymes, contributing to NSS formation.

Lines 134-137: Where it reads: “However, this does not prevent uncoupling of eNOS and is associated with ventricular remodeling, which suggest that the presence and participation of uncoupled eNOS in cardiomyocytes and/or fibroblasts in cardiac hypertrophy and can be considered as the missing link between atrial fibrillation and chronic non-ischemic cardiomyopathy”, it should read: However, this does not prevent uncoupling of eNOS and is associated with ventricular remodeling, which suggest the presence and participation of uncoupled eNOS in cardiomyocytes and/or fibroblasts in cardiac hypertrophy which can be considered as the missing link between atrial fibrillation and chronic non-ischemic cardiomyopathy.

Lines 176-178: Where it reads:” The Figure 2 describes the relation of the NSS with some metabolic disorders that which addresses the review”, it should read: Figure 2 describes the relationship between NSS and metabolic disorders which will be addressed in this review.

Lines 185-186: Where it reads: “However, only a small percentage of cysteine is susceptible to modification by ONOO– [56].”, it should read: However, only a small percentage of cysteine residues within proteins are susceptible to modification by ONOO–. The authors should explain what characteristics make a cysteine residue susceptible to ONOO- oxidation (reactive cysteine).

Lines 206-207: The authors should develop/explain the following, for instance by giving specific examples: “The mitochondria play a critical role as antioxidant defences”.

Line 224: Where it reads: “Tyr nitration is the major mechanisms responsible for the irreversible inhibition of complex I provoked by adding exogenous NO that increase ONOO– levels”, it should read: Tyr nitration is the major mechanism responsible for the irreversible inhibition of mitochondrial complex I. This occurs via addition of exogenous NO leading to increases in ONOO– levels. The word: “provoked” is not appropriate in this context.

Line 261: Where it reads: “The PARP pharmacological inhibition or genetic deletion of PARP-1 preserves cellular NAD+”, it should read: PARP-1 pharmacological inhibition or genetic deletion preserves cellular NAD+.

Lines 262-263: “ATP pools in oxidatively and/or nitrosatively stressed endothelial cells (as well as many other cell types), thereby allowing them to function normally.” It would be more appropriate: ATP pools in oxidatively and/or nitrosatively stressed cells, thereby … References are missing at the end of this sentence.

Line 288: Where it reads: “On another hand”, it should be: On the other hand. However I do not think that this is appropriate in the context of the text, and in fact should be removed.

Line 291: “This last isoform has a molecular mass of 30kDa”. I do not understand why it is relevant to put the molecular weight of Cu/Zn-SOD. I suggest removing this information.

Line 312: Where it reads: “The glutathione peroxidase (GPx) isoform family”, it should read: The glutathione peroxidase (GPx) family.

Lines 313-314: Where it reads: “The GPx1 is the main antioxidant enzyme preventing the accumulation of damaging intracellular by H2O2, it uses GSH and belongs to the GPxs family”, it should read: The GPx1 is the main antioxidant enzyme preventing the accumulation of intracellular damage by H2O2, in a GSH dependent manner.

“and belongs to the GPxs family” – this should be removed as it is already mentioned at the beginning of the paragraph.

Lines 319-321: “In the case of 2-Cys peroxiredoxin, small oxide-reductases with redox centers comprising two Cys-residues regenerate the catalytic site, whereas a cysteine residue in GPx is reduced by GSH [103].” This sentence needs thorough editing of the English language for clarification of its content.

Line 365: Where it reads: “that acts in the control the Zn2+ concentration”, it should read: “that acts in the control of Zn2+ concentration(s)”.

Line 372: Where it reads: “resulting in a vicious circle”, it should be: resulting in a vicious cycle.

Lines 382-386: The following sentence needs thorough editing of the English language: “Several reports have concluded that Zn2+ functions as a cyto-protective agent, defending cells

383 against oxidative insults and stimuli that induce apoptosis. This protective effect includes serving as a factor in the maintenance and reparation of the cellular membranes activation of anti-apoptotic signal transduction [128], inhibition pro-apoptotic of the NFĸB, activator protein 1 (AP-1) [129], antagonism of LPO and inhibition of caspases and endonucleases [130].”

Line 435: Where it reads: “neovascularization and macular edema and can also occur”, it should read: neovascularization and macular edema can also occur.

Lines 459-463: “Eales disease is an idiopathic inflammatory venous occlusion condition with male predominance (99%) that affects the peripheral retina of young adults (15 to 45 years). Inflammation is the main clinical feature of this pathology [159], where there is also an increases the aldose reductase activity, a decreases of the Na/KATP-ase activity [160] and an imbalance in the NAD+/NADH and NADP+/NAD(P)H redox levels. These changes are associated with NSS [161].” The authors should explain in more detail how NSS induces/is associated with the changes mentioned above.

Line 492: Where it reads: “NSS plays an important role in the pathogenesis of CVD”, it should read: NSS plays an important role in the pathogenesis of cardiovascular diseases (CVD).

Lines 558-560: The following sentence needs editing of the English language: “The regulatory roles of NO and ONOO– by endothelial cells are controlled by their production rates and their concentrations and effect on vascular smooth muscle cells (VSMC) [199].”

Line 564: “SOD inactivation also increases NSS and induces tissue damage [202].” The authors should develop this theme and give some examples.

Lines 579-581: The following sentence needs editing of the English language: “The main NO receptor initiating the canonical signaling pathway is the sGC receptor which is responsible for the enzymatic e conversion of GTP into cGMP, and is crucial for the modulation of the tone vascular.”

Line 604: The English language in the section: “11.2. NSS and essential arterial hypertension and metabolic syndrome-related hypertension” is particularly problematic (especially in the first 4-5 paragraphs) and should be thoroughly examined. I would suggest shorter sentences in this section.

Lines 748-749: Where it reads: “ADMA can be transported into the cells through the inducible L-arg transporter CAT-2B, that is the same transporter that uses the L-arg.”, it should read: ADMA can be transported into the cells through the inducible L-arg transporter CAT-2B, that is the same transporter for L-arg. Alternatively, the last part of the sentence “that is the same transporter for L-arg” should be removed, because it is redundant.

Lines 754-755: “and result in a hypoxic response” – the authors should explain what is hypoxia?

Line 756: “Activated HIF-1α decreases mitochondrial biogenesis by the HIF-1α–PGC-1α–TFAM axis” – what is PGC and TFAM? The authors use only the abbreviations for these proteins.

Lines 762-763: The following sentence is confusing: “Moreover, the exogenous administration of inhaled NO can alter the endogenous pulmonary endothelial function [278].” Do the authors mean: The inhalation of NO can alter … please clarify.

Lines 762-772: This paragraph needs revision of the English language.

Line 781: “During severe sepsis” – the authors should add the definition of sepsis.

Line 803: In the section “13. NSS and cancer”, the authors show a focus on prostate cancer and very little detail regarding other types of cancer. This appears unbalanced. Is there more studies focus on NSS in prostate cancer? Otherwise the authors should develop this section including studies focused in other types of cancer with more detail.

Suggestion: Figure with diagrams for the different NOS.

Author Response

(The authors gave the same response as above.)

Reviewer 3 Report

This is a comprehensive, detailed and multi-leveled coverage of the different biopharmaceutical implications of nitric oxide stress in diverse diseases, tissues and processes. The topic is of considerable importance, which justifies the assembly of new information into a coherent entity which can be of use to many and is a strength point of this review.  However, this topic had already been covered by many for a number of years, such that publishing a new review on this issue calls for a new point of view, a fresh concept or a link between the impact of nitric oxide stress in different tissues and organs. Unfortunately, I have failed to find such a fresh line of thought or any novel link between the different sub-topics covered in this lengthy text, which is a major point of weakness in this manuscript and calls for a substantial revision, as is listed below.

  1. Most importantly, no critical thinking is seen throughout this long text, which simply lists all of those instances where NO stress may exert some damage, without comparing the severity of the listed phenomena or explaining the order in which those are presented.
  2. In particular, body-brain communication processes and their role as transmitting the message of impaired NO pathways are missing. Specifically, the links between NO imbalance and muscarinic regulation in arteries should be covered.
  3. Further, the great majority of the human genome is devoted to long and short non-coding transcripts which nevertheless have key biological roles, but this review totally ignores that aspect of human biology which is badly absent here. Examples include microRNA regulators of inflammation such as miR-132 and its impact on NO stress.
  4. Also, the text is far too long for the ordinary scientist to go through. Try to cut the length of this text by half, which further will enforce focusing on the more important issues.
  5. Adding a table or two where the less central phenomena can be listed may be of assistance in this task of shortening this lengthy review.
  6. Add clear and informative 3-5 illustrations highlighting the main messages, in the form of a graphical abstracts which will open each section. Those can be of major importance and will also aid in selecting the topics of interest in this long text.
  7. Additionally, the bibliography is excessively long and it appears as if the citations were not filtered at all, such that one finds numerous citations of negligible journals and lacks the capacity to appreciate which of the citations is more important or worthy. To correct this flaw, the bibliography should be sharply cut, with those journals of higher impact factors preferred as they may be considered to have been subjected to critical reviewing by professional experts; and newer references might be preferred considering that such reviews might be more-timely than its present version.

Author Response

(The authors gave the same response as above.)

Round 2

Reviewer 1 Report

The authors have addressed minor concerns pointed out by reviewers, but did not address the major weakness of this manuscript – lack of novel points of view and overly lengthy content that has no strong focus.

Reviewer 3 Report

My opinion on this review had not changed, although i appreciate the work invested in revising it.  One critical demand is to cut the bibliography to at most 100 references and pay attention to the value of the journals where the cited references appeared.